# Contribution of Predictive and Prognostic Biomarkers to Clinical Research on Chronic Kidney Disease

**DOI:** 10.3390/ijms21165846

**Published:** 2020-08-14

**Authors:** Michele Provenzano, Salvatore Rotundo, Paolo Chiodini, Ida Gagliardi, Ashour Michael, Elvira Angotti, Silvio Borrelli, Raffaele Serra, Daniela Foti, Giovambattista De Sarro, Michele Andreucci

**Affiliations:** 1Renal Unit, Department of Health Sciences, “Magna Graecia” University of Catanzaro, I-88100 Catanzaro, Italy; ida_88@libero.it (I.G.); ashourmichael@yahoo.com (A.M.); 2Department of Health Sciences, “Magna Graecia” University of Catanzaro, I-88100 Catanzaro, Italy; srotundo91@gmail.com (S.R.); foti@unicz.it (D.F.); 3Medical Statistics Unit, University of Campania Luigi Vanvitelli, I-80138 Naples, Italy; paolo.chiodini@unicampania.it; 4Clinical Biochemistry Unit, Azienda Ospedaliera Universitaria Mater Domini Hospital, I-88100 Catanzaro, Italy; e.angotti@materdominiaou.it; 5Renal Unit, University of Campania “Luigi Vanvitelli”, I-80138 Naples, Italy; dott.silvioborrelli@gmail.com; 6Interuniversity Center of Phlebolymphology (CIFL), “Magna Graecia” University of Catanzaro, I-88100 Catanzaro, Italy; rserra@unicz.it; 7Pharmacology Unit, Department of Health Sciences, School of Medicine, “Magna Graecia” University of Catanzaro, I-88100 Catanzaro, Italy; desarro@unicz.it

**Keywords:** end-stage kidney disease (ESKD), cardiovascular disease, epidemiology, CKD, biomarkers

## Abstract

Chronic kidney disease (CKD), defined as the presence of albuminuria and/or reduction in estimated glomerular filtration rate (eGFR) < 60 mL/min/1.73 m^2^, is considered a growing public health problem, with its prevalence and incidence having almost doubled in the past three decades. The implementation of novel biomarkers in clinical practice is crucial, since it could allow earlier diagnosis and lead to an improvement in CKD outcomes. Nevertheless, a clear guidance on how to develop biomarkers in the setting of CKD is not yet available. The aim of this review is to report the framework for implementing biomarkers in observational and intervention studies. Biomarkers are classified as either prognostic or predictive; the first type is used to identify the likelihood of a patient to develop an endpoint regardless of treatment, whereas the second type is used to determine whether the patient is likely to benefit from a specific treatment. Many single assays and complex biomarkers were shown to improve the prediction of cardiovascular and kidney outcomes in CKD patients on top of the traditional risk factors. Biomarkers were also shown to improve clinical trial designs. Understanding the correct ways to validate and implement novel biomarkers in CKD will help to mitigate the global burden of CKD and to improve the individual prognosis of these high-risk patients.

## 1. Introduction

A biomarker is defined, by a collaborative working group involved with both the United States National Institutes of Health (NIH) and the Food and Drug Administration (FDA), as “a characteristic that is measured as an indicator of normal biological processes, pathogenic processes, or responses to an exposure or intervention, including therapeutic interventions” [1]. This working group definition was formed on the initiative of the NIH with the aim of accelerating the development and clinical application of reliable biomarkers based on shared definitions. Indeed, an “ideal” biomarker is defined with the presence of some analytic features: (1) it should be measured and readily available in biological samples, such as blood or urine; (2) it should be reproducible, non-invasive, and not expensive [2]. In addition, several clinical features should also be provided to complete the biomarker’s definition; it needs to allow for an early detection of a disease status, while it also needs to have high sensitivity and specificity, i.e., the biomarker needs to differentiate the pathologic status from the normal one and from other clinical conditions, as accurately as possible [3]. The effort made by the NIH–FDA working group is considerable ever since it forecasted and tried to solve the problems of biomarker development, from discovery to clinical application. Indeed, once a biomarker is found to be involved in one or more pathophysiological mechanisms of a disease, it may be introduced into clinical practice to see whether it offers advantages in clinical management and after completion of the validation phase that is considered a crucial step [4]. An important example of the pitfalls of biomarker development is illustrated in the chronic kidney disease (CKD) scenario. CKD is a chronic disease characterized by a poor prognosis, due to the strong association with the development of cardiovascular events, all-cause mortality, and renal events such as renal replacement therapies (RRT, i.e., dialysis or kidney transplantation) [5,6]. The Kidney Disease Improving Global Outcomes Work Group (KDIGO), in 2012, defined CKD with the presence of either decreased kidney function (estimated glomerular filtration rate (eGFR) < 60 mL/min/1.73 m^2^) and/or albuminuria, namely, an abnormal amount of protein excretion with urine, for at least three months [7]. The global dimension of the disease is so important that CKD started to be considered a relevant public health problem. Indeed, according to the Global Burden of Kidney Disease, the CKD incidence and prevalence increased by 88.76% and 86.95%, respectively, from 1990 to 2016 [8]. Moreover, the mortality attributed to CKD increased by 41.5% between 1990 and 2017, a percentage that exceeded the mortality due to several neoplasms or cardiovascular (CV) disease [9]. Hence, great effort is advocated toward improving clinical decision-making and reinforcing treatment and prevention of CKD. The KDIGO working group proposed a classification called “CGA” that incorporates the cause (C) of kidney disease, as well as the eGFR (G) and albuminuria (A) levels, to stratify risk in patients with CKD and to better address the importance of the underlying disease. However, considering eGFR and albuminuria levels in cross-tabs, like those reported in KDIGO guidelines, does not take into consideration the complex mechanisms of CKD. In fact, all patients are stratified on the basis of eGFR and albuminuria and allocated in risk categories for prognostic estimations. This approach was also defined as “reductionist”, since it does not consider the different etiologies of CKD and many other parameters, including serum or urine biomarkers, which could help clinical management of CKD [10]. Moreover, although the KDIGO also suggested considering the causes of kidney disease to predict poor outcomes in CKD patients, how to incorporate these parameters remains unclear [11]. Few studies included renal diagnoses in risk prediction models and, when diagnoses are present, they are classified in different ways (i.e., four, five, or even six categories) with large heterogeneity between studies, thus making an univocal interpretation difficult [12,13]. There is also a growing debate regarding the question whether an eGFR reduction below the threshold of 60 mL/min/1.73 m^2^ represents the consequence of a physiologic senescence or a marker of renal pathology [14,15]. In fact, despite the evidence that eGFR reduction foresees both the onset of end-stage kidney disease (ESKD) and the all-cause death regardless of age, in large general and high-risk populations, other human studies showed that kidneys undergo structural and functional change with aging such as nephrosclerosis and a reduction in measured GFR [14,16]. At the same time, these changes are not associated with a reduction in single-nephron GFR; this is the likely reason why, in the absence of albuminuria, the GFR reduction alone determines only a small increased risk for age-standardized mortality and ESKD [16]. Owing to these important controversies and in an attempt to improve risk stratification and care of CKD patients, as well as to share clinical findings with the nephrology community, the International Society of Nephrology (ISN) started an international project called “closing the gaps”, which encompasses a set of activities that have to be performed to improve the prognosis of CKD patients [17]. The core of this project is about the development, implementation, and clinical application of novel biomarkers in CKD patients. It indeed emerged that, with the exception of cystatin C, which is a filtration marker useful to estimate the eGFR and to improve risk prediction in CKD, the vast majority of previously tested biomarkers in CKD did not reach any clinical application [18]. The aim of this review is to report the principal pieces of evidence regarding biomarker development in CKD, the contribution of these biomarkers to both observational and interventional (i.e., randomized controlled trials) studies, and the possible strategies that could be followed to ameliorate this important branch of clinical research.

## 2. General Classification of Biomarkers 

Depending on the intended use, biomarkers are classified as diagnostic, pharmacodynamic/response, monitoring, prognostic, or predictive [19]. We focus, in the present review article, on prognostic and predictive biomarkers, as they represent the most developed biomarkers in CKD patients, while they also incorporate characteristics from other categories of biomarkers. A prognostic biomarker is used to identify the probability of a clinical outcome in patients who are already suffering from the disease of interest [1,20]. Furthermore, prognostic biomarkers measure the association between the disease and clinical outcome in the absence of therapy or with standard therapy that all patients are likely to receive. On the other hand, predictive biomarkers are used to determine whether a patient is likely to benefit from a particular therapy. The clinical benefit could be either a good response to a drug if the biomarker is positive or, alternatively, a lack of benefit from the same drug, which can save a patient from drug toxicity or unnecessary side effects [1].

## 3. Prognostic Biomarkers in CKD

The importance of prognostic biomarkers in the CKD setting is crucial. Indeed, CKD is a multifactorial disease in which risk factors play different roles in different individuals and in different stages of the disease. It was demonstrated, for example, that the presence of type 2 diabetes leads to the development of CKD in up to 30% of subjects. This means that, in these patients, the deleterious pathogenetic mechanisms of diabetes mellitus are sufficient to damage the kidneys with the onset of typical diabetic glomerulosclerosis. Conversely, it is also possible that, in a portion of diabetic patients, the kidneys are injured by the co-existence of arterial hypertension, which causes different lesions (mainly injuring the kidney vessels), with a completely different prognosis. 

### 3.1. Kidney Biomarkers

The assessment of correct risk stratification, i.e., the allocation of CKD patients to the true risk-of-event categories, always represents a difficult challenge for nephrologists and researchers, due to the large variability in etiology and prognosis of CKD [21,22]. In order to accomplish this aim, a growing number of risk prediction models in CKD patients were developed over the past several years. They show how kidney measures, such as albuminuria (or proteinuria) and eGFR, are strong prognostic biomarkers. Indeed, an eGFR reduction to levels below 60 mL/min/1.73 m^2^ or even a small increase in albuminuria levels is associated with a significantly increased risk for CV events (CV mortality, coronary heart disease, stroke, heart failure), all-cause mortality, and ESKD (the most advanced stage of CKD that requires referral to renal replacement therapies such as hemodialysis), both in the general population and in patients with an already established CKD (Figure 1) [23,24,25].

Albuminuria and eGFR were also recently used to develop individual risk prediction models for both CV and renal risk in CKD patients. These models employ statistical measures (such as calibration, discrimination, and validation) that represent an essential step before applying risk estimates at the individual level [25,26,28,29]. Moreover, risk prediction models are available for patients with early, moderate, and severe stages of CKD. The addition of albuminuria and eGFR to traditional risk factors included in the model (such as age, gender, presence of diabetes, blood pressure, serum cholesterol) was associated with a significant improvement in risk prediction. Even more importantly, the contribution of albuminuria and eGFR to the prediction of CV events (CV mortality, coronary heart disease, stroke, and heart failure) was found to be greater than traditional CV risk factors [24]. This notwithstanding, a main limitation of available risk score is the insufficient, even absent, consideration of underlying causes of renal disease, which is a crucial point when considering that CKD is a set of multiple etiologies rather than a single disease [12,13]. However, these first prediction models established that measuring albuminuria and eGFR is a central step for assessing risk stratification in CKD patients. Mechanisms of damage of albuminuria and eGFR were partially explained. Albuminuria was shown to exert a direct harmful effect on renal glomeruli and tubules [30]. Moreover, it can also be considered as a systemic marker of endothelial dysfunction, and this explains the reason for which the presence of proteinuria strictly forecasts the onset of CV events in the general population, as well as CKD patients [31]. Similarly, eGFR reduction is linked to an increase of uremic toxins that are responsible for kidney and systemic damage. A drop in eGFR was also associated with the development of coronary atherosclerosis, regardless of the presence of diabetes mellitus, dyslipidemia, previous CV disease, and other comorbidities [32]. Several studies showed that an eGFR decrease is associated with the onset of sudden cardiac death (SCD), with the data being confirmed from the early stage of CKD, i.e., from eGFR < 60 mL/min/1.73 m^2^ [33,34,35]. For each 10 mL/min decrement in eGFR, SCD risk increased by 11% [33]. SCD accounts for the vast majority of deaths in CKD patients and is also mediated by metabolic and electrophysiological abnormalities [36]. However, both eGFR and albuminuria have limitations in risk prediction. Albuminuria is not specific for any kidney disease, occurring in ischemic, diabetic, and tubulointerstitial nephropathies, as well as in the vast majority of glomerulonephritis and autoimmune diseases [13]. Albuminuria also presents a random variability, since urine protein excretion follows a circadian rhythm that is influenced by posture, exercise, or dietary factors [37]. The eGFR, albeit associated with a poor prognosis, could also be altered by temporary or reversible clinical conditions, such as volume depletions or sub-acute tubulointerstitial diseases [11]. Taken together, proteinuria and eGFR share the limitation of detecting kidney damage often when it is already established and is, thus, not reversible [15]. Furthermore, urine protein excretion is strictly dependent on eGFR levels, which are an expression of the number of functional nephrons in the kidney. In fact, we recently demonstrated that, in a model in which proteinuria is replaced by F-Uprot (proteinuria/eGFR × 100), an expression of the combination of the two biomarkers, the latter allowed refining risk stratification for ESKD outcome in all CKD stages, even in more advanced CKD [27]. Albuminuria levels are strictly dependent and, thus, modified by both systolic and diastolic blood pressure (BP). As long as the eGFR decreases, a given increase in BP is accompanied by an increase in urine protein excretion. This phenomenon was observed in both animal and human studies and is caused by the “remnant nephron effect”, namely, the transmission of systemic hydrostatic pressure to the glomerular microcirculation [38,39]. Moreover, BP is a clinical parameter characterized per se by high variability, which is also detectable during 24-h ambulatory BP measurements (short-term variability) [40]. It was demonstrated that both systolic and diastolic BP variability influence albuminuria levels [41]. Hence, a number of variables were shown to influence albuminuria levels, and this could lead to biased risk estimation, particularly if the risk prediction is based on a single measurement of albuminuria. The ISN indeed highlighted that a consensus needs to be found on how often albuminuria should be measured to warrant a true prediction of cardiovascular and renal endpoints, as well as to monitor the course of CKD [18]. Owing to the evidence that eGFR and albuminuria are able to provide a strong but incomplete prediction of cardiorenal endpoints in CKD patients, the next step of prognostic research focused on the development and assessment of biomarkers that provide useful prognostic information beyond proteinuria and eGFR. A number of markers of inflammation, oxidative stress, or tissue remodeling aroused interest in improving CV and renal risk prediction in CKD patients. 

### 3.2. Markers of Oxidative Stress, Tissue Remodeling, and Metabolism 

Myeloperoxidase (MPO) is a biomarker of oxidative stress that fosters nitic oxide consumption and which is associated with the development of atherosclerotic lesions, CV disease, and eGFR decline in CKD [42,43]. A recent observational analysis of the Chronic Renal Insufficiency Cohort (CRIC), which enrolled approximately 4000 patients with CKD in the United States (US), showed that serum MPO levels were associated with the risk of renal outcome, defined as initiation of RRT, 50% eGFR decline, or eGFR ≤ 15 mL/min/1.73 m^2^ [44]. The key element of this analysis was that the effect of MPO was significant even after adjustment for main confounders, such as baseline eGFR and proteinuria levels. Matrix metalloproteinases (MMPs), endopeptidases involved in tissue development, and homeostasis through the regulation of cell differentiation, apoptosis, and angiogenesis were shown to intervene in inflammatory and fibrotic processes across the kidneys [45,46]. Blood and urine levels of MMPs were linked to renal and CV disease in previous clinical studies in humans. Serum and urine MMP-2, -8, and -9 levels are increased in patients with diabetic CKD, with MMP-9 being significantly associated with the severity of albuminuria [47,48]. Increased plasma levels of TIMP-1 (tissue inhibitor of metalloproteinases-1) predicted the incidence of CKD regardless of inflammatory markers such as C-reactive protein [49]. MMPs and TIMPs also play a role in accelerating the atherosclerotic process by increasing cell migration to the plaque fibrous cap that in turn determines plaque inflammation and rupture [50]. Indeed, the levels of several MMPs (MMP-1, -2, -8, -9) and TIMP-1 were found to be increased in patients with peripheral arterial disease, including those with aneurysms of the arterial wall [51]. Fibroblast growth factor-23 (FGF-23), a hormone involved in phosphorus metabolism that increases progressively as kidney function declines, was significantly associated with mortality, atherosclerotic events, heart failure (HF), and ESKD in CKD patients [52,53].

### 3.3. Cardiac Biomarkers

Several cardiac biomarkers were investigated, mainly for establishing their role in CV and renal risk prediction in CKD patients. Cardiac troponins (high-sensitivity cardiac troponin (hs-cTnT)) and natriuretic peptides (N-terminal pro-B-type natriuretic peptide (NT-proBNP)) are largely used in CV medicine to diagnose coronary artery disease (CAD) and heart failure (HF), respectively. Both these biomarkers are, thus, an expression of subclinical abnormalities in the heart [54,55]. However, one major problem that makes their introduction in individual risk prediction difficult is that cardiac markers are an expression of both cardiac and kidney dysfunction and cannot discern these two conditions. Natriuretic peptides act by promoting the tubular natriuresis across the kidney and counteracting the effects of renin–angiotensin–aldosterone system, which is triggered by heart failure, as well as renal dysfunction [56]. Concerns about the interpretation of hs-cTnT and natriuretic peptides also derive from the evidence that these marker concentrations are influenced by kidney function levels [56,57]. So far, cardiac markers found more application in the context of prognostic estimation of cardiorenal syndromes (CRS), clinical disorders where an acute or chronic dysfunction of one organ may lead to an acute or chronic dysfunction of the other, thus testifying the strict relationship between the heart and the kidney [58]. In the general population, hs-cTnT and NT-proBNP were shown to be strong predictors for incident HF over time [59,60]. In the setting of CKD, an attempt to evaluate, with appropriate statistical tools, the contribution of cardiac markers to the development of CV events was made using the Atherosclerosis Risk in Communities study (ARIC) population. In this study, examining 7682 non-CKD and 970 patients with CKD stage 1–5, hs-cTnT and NT-proBNP were associated with the development of CV events (defined as the composite of coronary heart disease, stroke, and HF) independently of kidney measures (eGFR and albuminuria) [61]. The finding was confirmed for both CKD and non-CKD patients, as well as for patients with or without previous CV disease. However, the interpretation of these results should be done with caution since the ARIC cohort was stratified, for this analysis, by the presence/absence of CKD, thus limiting the influence of kidney measures on CV risk prediction [61]. Results of the association between cardiac markers and renal outcomes in CKD patients are even more conflicting. The CRIC investigators found, in a cohort of over 3000 CKD patients, that increased plasma levels of growth differentiation factor-15 (GDF-15, a member of the transforming growth factor (TGF)-β cytokine family), hs-cTnT, and NT-proBNP were associated with CKD progression. defined as the onset of ESKD or 50% eGFR decline [62]. However, when all these parameters were added to the prediction model including traditional CV risk factors, the model discrimination (i.e., the ability of the model to separate individuals who develop events from those who do not; see more details in Section 5) did not improve, meaning that their clinical utility was scarce. Moreover, in the Framingham cohort, hs-cTnT was not associated with a faster eGFR decline or with incident CKD [63]. There is, overall, a need for future work to assess the role of cardiac markers in CV and renal risk prediction [61,62,63].

### 3.4. Filtration and Urinary Biomarkers 

Filtration biomarkers and urinary markers were also investigated. The use of cystatin C to estimate GFR (eGFR_cys_) improved the risk stratification for death, death from CV causes, and ESKD with a large proportion (23%) of patients being reclassified toward true risk estimates when compared with eGFR estimated from serum creatinine (eGFR_crea_) [64]. eGFR_cys_ was also shown to predict the onset of SCD in elderly CKD patients [35]. The combination of serum creatinine and cystatin C for estimating eGFR (eGFR_cys-crea_) allowed clinicians to anticipate the risk prediction of worse outcomes at 85 mL/min, which is well above the 60 mL/min threshold defined by eGFR_crea_. β2-microglobulin, another filtration marker, showed a statistical power similar to cystatin C in improving prediction of ESKD, all-cause mortality, and new onset of CV disease beyond eGFR_crea_ [65]. With respect to urinary markers, some evidence, albeit controversial, was provided for the association of urinary markers of tubule damage (interleukin (IL)-18, kidney injury molecule-1 (KIM-1), neutrophil gelatinase-associated lipocalin (NGAL)), repair (human cartilage glycoprotein-40 (YKL-40)), and inflammation (monocyte chemoattractant protein-1 (MCP-1)) with the risk of ESKD [66,67]. In fact, when risk prediction models were adjusted for baseline eGFR and albuminuria, the associations of these biomarkers with clinical outcome were consistently attenuated. However, the highest values of KIM-1, MCP-1, and YKL-40 also provided useful risk estimation beyond eGFR and albuminuria in a post hoc analysis of the Systolic Blood Pressure Intervention Trial SPRINT trial [68]. Interestingly, urinary IL-18 and NGAL levels were shown to predict linear eGFR decline over time, an endpoint of growing interest in clinical research [68].

### 3.5. Prognostic Role of Proteomics, Metabolomics, and Genomics

Proteomics metabolomics and genomics recently provided great input to the implementation of novel biomarkers [69,70,71]. The advantage of these “omics” techniques is to provide a combination of informative peptides/metabolites that are able to classify patients (hence, the appellation of classifiers) into significant clinical or risk categories. A well-depicted classifier in CKD patients is the CKD273, a panel of 273 urine peptides shown to predict, in long-term follow-up cohort studies, eGFR decline with a strong and independent effect to the onset of albuminuria, particularly in diabetic patients [72,73,74]. In further risk prediction models, CKD273 was also able to reclassify about 30% of patients compared with the standard equation that considers eGFR and albuminuria, for the risk of CKD progression [75]. The CRIC investigators described, in a recent manuscript, the association between a panel of 13 urine metabolites and CKD progression [76]. Results of this analysis are encouraging since the levels of four metabolites, namely, 3-hydroxyisobutyrate (3-HIBA), 3-methylcrotonyglycine, citric acid, and aconitic acid, were associated with eGFR decline, with 3-HIBA and aconitic acid levels also significantly associated with the hard endpoint ESKD. Of particular interest is the prognostic role of the genetic causes of CKD. Of all CKD cases diagnosed at a young age (<25 years), an actual 30% are determined by monogenic disorders, and inherited CKD is globally more prevalent (prevalence ranged between 30% and 75%) than previously thought, particularly in the presence of a family history of CKD [77,78]. More importantly, the advent of genome-wide association studies (GWAS) allowed the discovery of several single-nucleotide polymorphisms (SNPs) associated with an increased risk for CKD or with a worse prognosis in patients already affected by CKD [78]. Polymorphisms in the Uromodulin (UMOD) gene region rs4293393, which codifies the most abundant urinary protein in healthy subjects, namely, uromodulin (also called Tamm–Horsfall protein), are associated with an increased risk of incident CKD [79]. A similar role in predicting the onset of CKD was exerted by other SNPs such as Protein Kinase AMP-Activated Non-Catalytic Subunit Gamma 2 (PRKAG2), Longevity Assurance Gene Homologs (LASS2), Disabled Homolog 2 (alias DAB Adaptor Protein 2, DAB2), Dachshund Family Transcription Factor 1 (DACH1), and Stanniocalcin 1 (STC1) [80]. Apolipoprotein L1 (APOL1) gene variants were also studied in CKD patients. APOL1 encodes apolipoprotein L1, which is involved in the lysis of *Trypanosoma brucei* and other trypanosomes [81]. The G1 and G2 variants of APOL1 were associated with an increased risk of eGFR decline and disease progression to ESKD in CKD populations [82]. Interestingly, information derived from the SNPs were recently combined into a genetic risk score [78]. This score was shown to be associated with eGFR decline and kidney outcome regardless of albuminuria and other renal risk factors encompassing diabetes, history of CV disease, and hypertension. A number of studies assessing the associations between SNPs and kidney measures were carried-out by the United Kingdom (UK) biobank, a large cohort of over 500,000 participants enrolled in 2006–2010, from which genotypic information was widely collected [83]. Analyses of the UK biobank provided a great contribution to the prognostic research in CKD. For example, genetically predicted testosterone and fasting insulin, with the latter being an expression of insulin resistance, were found to be associated with CKD and worse kidney function in men, thus highlighting the possible reasons for discrepancy in CKD prevalence and CKD progression among men and women [84,85]. Intriguingly, a genome-wide association study of UK biobank showed that albumin-to-creatinine ratio (ACR) is dependent on multiple pathways and that an ACR genetic risk score may improve the prediction of hypertension and stroke [86].

## 4. Predictive Biomarkers in CKD

Predictive biomarkers are used in disparate fields of medicine to assess the likelihood of response to treatments and the individual pathophysiology of the disease. One major example of this strategy is represented by the large use of predictive biomarkers in oncology. Causative mutations of the breast cancer genes 1 and 2 (BRCA1/2) were found to be predictive biomarkers for identifying the response to poly(ADP-ribose) polymerase (PARP) inhibitors [87]. Such a discovery is crucial as BRCA1/2 provide information on the best drug for the individual patient in order to improve their prognosis. While, in oncology, a set of pathophysiological mechanisms is crucial for tumor development, what complicates the application of predictive biomarkers in chronic diseases is that different mechanisms are active in different stages of the disease itself and in different patients [88]. This means that, if a treatment is started on the basis of a blood/urine biomarker level, the individual prognosis may remain unchanged or even worsen, due to the presence of other active mechanisms of damage, as well as, most importantly, different disease entities that cause the chronic decline of renal function through diverse pathophysiological pathways. Notwithstanding, in chronic disease, great research effort was also started with the aim of personalizing treatments following the methodological concept of “the right drug for the right patient”. Hence, the implementation of predictive biomarkers represents a topic of increasing importance.

### 4.1. Kidney Biomarkers

In nephrology, the most used predictive biomarkers are eGFR and albuminuria. Both these biomarkers can be considered as “dynamic” predictive biomarkers. In fact, their levels change over time with the effects of treatment, such that they can be efficiently used for monitoring the course of CKD and the appropriateness of the therapy followed by the patient. In the past few decades, several interventional studies were carried out testing the effect of nephroprotective drugs on hard endpoints such as mortality, CV events, and ESKD in patients with CKD [89,90,91,92,93,94,95,96]. Although interventions differed between studies, with principally antihypertensive drugs and albuminuria-lowering agents being tested, all these trials pointed out that the CV, mortality, and ESKD risk reductions were strictly associated with a reduction in albuminuria after the start of treatment. Moreover, the magnitude of treatment effect was greater in patients with higher albuminuria levels at the time of the initial visit [95,96]. These findings are reinforced by the evidence that albuminuria changes also played a potentially beneficial role in negative clinical trials. In the Aliskiren Trial in Type 2 Diabetes Using Cardiorenal Endpoints (ALTITUDE), which failed in demonstrating the advantage of adding Aliskiren to an angiotensin-converting enzyme inhibitor (ACEi) or an angiotensin-receptor blocker (ARB) on CV and renal outcomes, patients who showed an albuminuria reduction in the Aliskiren arm (37%) were largely protected against CKD progression compared with those who did not show a reduction in albuminuria levels [97]. All these pieces of evidence testified that albuminuria has great predictive and prognostic power in CKD patients and, although further studies are needed to find the correct threshold of albuminuria reduction that can confer CV and renal risk protection after an appropriate treatment, there is a general consensus that a 30% reduction in its levels from baseline to six months could be acceptable [98]. With respect to eGFR, it was demonstrated that a doubling of serum creatinine level, which corresponds approximately to a 57% eGFR decline, was able to predict CKD progression in previous clinical trials in diabetic CKD patients [99]. The importance of that evidence is highlighted by the fact that, in these previous trials, eGFR decline correlated with renal outcomes after exposure to nephroprotective treatments, thus affirming its role as a predictive biomarker, in addition to a prognostic biomarker [100]. Since then, the association of lesser eGFR declines with CKD outcomes was tested. A post hoc analysis of the Reduction of End Points in Non-Insulin-Dependent Diabetes with the Angiotensin II Antagonist Losartan (RENAAL) and Irbesartan Diabetic Nephropathy Trial (IDNT) clinical trials, two studies that evaluated the efficacy of ARB treatment in patients with diabetes mellitus and nephropathy, showed that 30% and 40% eGFR declines may improve the power of clinical trials if the drug investigated does not determine an acute (within three months of the start of treatment) drop in eGFR [101]. A larger meta-analysis of 37 clinical trials in CKD patients documented a strong association between 30% and 40% eGFR decline in the first 12 months of treatment and the onset of kidney disease progression [100]. 

### 4.2. Biomarkers of Tissue Remodeling

In addition to proteinuria and eGFR, other promising predictive biomarkers in CKD were described. A change in serum levels of MMPs after exposure to BB-1101, a synthetic hydroxamic acid-based inhibitor of MMP, was associated with a reduction in proteinuria in experimental models of glomerular damage [102]. A similar effect was observed in diabetic CKD patients who underwent treatment with doxycycline, an antibiotic from the tetracycline family, and who were already treated with renin–angiotensin–aldosterone inhibitors (RAAS-i) [103]. Moreover, MMPs are also involved in the mechanism that leads to CV risk reduction exerted by sodium–glucose cotransporter 2 inhibitors (SGLT2-i) through the activation of RECK (reversion-inducing cysteine-rich protein with kazal motifs), an endogenous inhibitor of MMPs [104]. That mechanism appeared to be independent of proteinuria levels and could also be useful for selecting high-risk normoalbuminuric CKD patients to be enrolled in future clinical trials, who represent a non-trivial proportion of the CKD cohort [21]. 

### 4.3. Ultrasound Biomarkers

Evidence is also emerging for a possible role of the renal resistive index (RRI) as a dynamic predictive biomarker. RRI is a Doppler ultrasonographic index, whose increase reflects both renal and systemic vascular impairment [105]. RRI was also found to predict the onset of CV and kidney outcomes in patients with CKD or essential hypertension [106,107]. RRI values are changed over time by different drug classes, such as RAAS-i and SGLT2-i; novel studies will hopefully reveal in the future if these treatment-induced modifications could also predict hard CV and renal endpoints [108,109]. 

### 4.4. Predictive Role of Proteomics, Metabolomics, and Genomics

A polymorphism of the angiotensin-converting enzyme gene caused by an insertion/deletion (ACE/ID) modifies the systemic and renal activity of the RAAS, which was recognized to be a trigger of kidney damage [110]. The ACE/DD–ACE/ID polymorphism was able to predict the response to losartan in type 2 diabetic patients enrolled in the RENAAL trial, that is, patients with worse prognosis (D allele carriers) had the best response to losartan [111]. Complex biomarkers and classifiers have a predictive role, in addition to a prognostic role. A set of 21 serum metabolites were selected from a larger panel through a penalized regression analysis, and they were shown to correctly predict the albuminuria response to ARB treatment in type 2 diabetic patients [112]. This classifier revealed that the enzyme nitric oxide synthase 3 (NOS3) is crucial to forecast the response to ARB therapy in diabetic CKD, since it is involved in the molecular mechanism of action of these drugs. A proteomic predictive classifier was developed from the Prevention of REnal and Vascular ENd-stage Disease (PREVEND) study, using plasma proteomics profiles of fibrosis and kidney damage that allowed predicting the albuminuria change in patients treated with RAAS-i [113]. The principal characteristics of prognostic and predictive biomarkers are depicted in Table 1.

## 5. Implementation of Biomarkers in Observational Studies

Owing to the importance of improving risk stratification beyond traditional kidney measures and to help clinical decision-making in CKD patients, the evaluation of novel biomarkers acquired great emphasis in clinical research, as witnessed by the growing number of publications on this topic [114]. However, before a biomarker can find full application in clinical practice, several steps need to be satisfied and reported. The first questions that should be addressed are the following: What will be the clinical intended use of the biomarker? Is the assay analytic performance acceptable for the intended use? To answer these important questions, the development process should start with assessing the analytic and clinical validity of the biomarker. Analytic validity refers to evaluating whether the characteristics of the measured biomarker are acceptable in term of precision, accuracy, and reproducibility [115]. It is indeed important to be aware that biomarker levels may vary in clinical practice due to factors not linked to the disease of interest being classified as pre-analytical and analytical factors [116]. Pre-analytical variation depends on several factors that include lifestyle (exercise, smoking habit, obesity), age, race, influence of gender, specimen collections (fasting, time, and temperature of storage) [117]. For instance, albuminuria, measured with the available methods, such as 24-h urine collection or albumin-to-creatinine ratio, is extremely influenced by physical exercise and other conditions that determine a day-by-day variation, defined as random variation [37]. Urinary NGAL concentration is stable in urine for up to seven days, but it is increased by the presence of white blood cells that are an important confounder [118]. Analytic variation is mainly defined by two parameters, which are bias and precision [119]. Bias is the amount by which an average of many repeated measurements made using the assay systematically over- or underestimates the true value. Precision represents the repeatability of measurements under unchanged assay conditions in a laboratory. While analytic validation is often discussed, it is seldom handled in a proper fashion. It was suggested to deepen analytic validation, while developing a biomarker, and to report metrics, such as precision, reproducibility, accuracy, analytic sensitivity, limits of detection and quantification, linearity, and analytic specificity [120]. A descriptive summary of these measures is reported in Table 2.

Clinical validity is the next important step and consists of demonstrating that biomarker measurement is associated with a clinical characteristic of interest [115]. The first steps of clinical validation are the proof of concept and prospective validation [114]. Proof of concept means to assess whether biomarker levels differ between subjects who develop the event of interest vs. non-events. This phase is essential since it allows understanding if the biomarker can play a role in the context of disease, and continuing its development is convenient. To this aim, a cross-sectional design could be sufficient [121]. Next, it is necessary, in prognostic validation, to evaluate if the biomarker is significantly associated with the event of interest, with a prospective analysis. Moreover, it is important that the magnitude of this association is not attenuated when the analysis is adjusted for traditional risk factors, such as age, gender, and proteinuria and eGFR levels in CKD patients. This step provides other useful information such as the distribution of the biomarker and, therefore, how to incorporate the biomarker levels in multivariable analyses. It is suggested to start by adding to the model the biomarker variable as a continuous variable, before applying a categorization (e.g., tertiles or quartiles) [122]. For instance, proteinuria has a skewed distribution and is often added as a log-transformed variable or restricted cubic spline in CKD prognostic models [27]. The prospective validation step is also important for selecting variables to be included in the model. This can be done by using a knowledge-driven (or a priori) method, based on the already known biological association of the variables with the outcome, or data-driven methods, which are automated tools that select a small set of variables from a larger one, in order to maximize the model fit [123]. In the case of a large number of predictors, as often happens during the development of proteomic/metabolomic classifiers, regularization or dimension reduction methods can be used [124]. The metabolomic classifier for the prediction of response to ARB treatment, which we described in Section 4, was developed by means of least absolute shrinkage and selection operator (LASSO), a regularization technique that shrinks the variables regression coefficients through a tuning parameter and retains the best predictors in the model. LASSO was also shown to work very well with small sample sizes [112]. The third phase of clinical validation is focused on the incremental value of the biomarker on the previous assessed risk models. In nephrology, what is essentially required in biomarker research is to demonstrate that a biomarker adds information, in the prediction of a defined endpoint, on top of already assessed risk factors. This process needs a hierarchical assessment, since a likelihood ratio test (LR test) should be firstly reported to determine if the biomarker remains associated with the endpoint after controlling for previously established risk factors. Next, three measures of performance should be reported: discrimination, calibration, and reclassification [125]. These three domains are important to warrant the applicability of the biomarker predictive performance to the individual patient. Discrimination refers to the ability of the model to attribute a high risk to patients who develop the outcome of interest and, accordingly, a low risk to those who do not [126]. A measure that depicts sensitivity and specificity for all possible thresholds of a biomarker is the receiver operating characteristic (ROC) curve. To evaluate discrimination, it is, thus, suggested to present the ROC derived from the model together with the area under the curve (AUC), also labeled the *c*-statistic [127]. The difference in *c*-statistic between models with and without the biomarker should also be presented. Calibration is the degree of agreement between observed and predicted outcomes. It is suggested to depict calibration graphically by plotting the mean predicted versus mean observed outcome probability for intervals (usually deciles) of risk in a predictiveness curve or by representing observed event rates versus mean predicted risk, thus creating a calibration plot, with points that should lie along a 45° line if the model is well calibrated [126]. Reclassification metrics provide useful information on the proportion (%) of patients that are reclassified in the true risk category (lower or higher risk), whether or not the new biomarker is added to a traditional risk prediction model. The most used reclassification metrics are the net reclassification improvement (NRI), the integrated discrimination index (IDI), and reclassification tables which directly depict the movement of patients between risk categories based on the risk predicted by models with and without the biomarker [127]. After showing measures of performance, it would be necessary to internally or externally validate the model. External validation implies that the risk prediction model, including the biomarker, is re-run within an external cohort of patients with similar characteristics (e.g., CKD patients) to confirm predictive accuracy in all sequences. Alternatively, several methods of internal validation, such as bootstrapping or cross-validation, can be computed [128]. The appropriateness of methodology used to develop a biomarker is a key element to obtain useful clinical results. This is particularly true if we consider that only a few prediction models in nephrology reported these measures appropriately [129]. However, this is not the only limitation. Most of the proposed biomarkers are yet to complete the sequence from discovery to clinical application, because they were developed in studies with a small sample size without validation, thus providing heterogeneous results. The ISN prompted that biomarker research would take advantage from the setting up of large, observational cohort studies and possibly a long-term follow-up in which biomarker development and validation could be strengthened and provide robust evidence for clinicians [18]. This also requires the standardization of data collection, storage, and database structure across countries, as well as a collaboration among academia, industry, and regulatory authorities in order to warrant a correct dissemination of results.

## 6. Biomarkers in Intervention Studies

Projecting clinical trials, which test the effect of novel pharmacological treatments on prognosis of CKD patients, is always an important challenge. In the past few decades, all nephrology communities expressed the need for clinicians to have more therapeutic tools, with each one specific for a particular etiology of CKD, in order to improve the care of patients with CKD and to deal promptly with the complexity of kidney disease, abandoning the “reductionist” approach [10,130]. The milestone of intervention studies in nephrology dates back to the years 1990–2000 when the Collaborative Study Group, the RENAAL, and the IDNT trials showed the efficacy of RAAS-i (ACEi and ARBs) in reducing CV and renal risk in patients with diabetes and CKD [76,131,132]. Since then, a number of clinical trials were carried out with an attempt to reduce the high residual risk in CKD patients, but they missed the target [133]. The reasons for this breakdown are several and include the enrolment, in clinical trials, of a large number of CKD patients with heterogeneous etiologies and the add-on strategy. The add-on strategy consists of adding a pharmacological agent to patients who are already being treated with a drug belonging to the same class. This was adopted, for example, in the Veterans Affairs Nephropathy in Diabetes (VA-NEPHRON-D) clinical trial, which tested the effect of dual RAAS blockade ACEi + ARB, or in the ALTITUDE trial, with the addition of Aliskiren, a renin inhibitor, to RAAS-i [134,135]. In these studies, the intensification of RAAS blockade did not result in further CV or renal risk protection and even increased the risk of these endpoints. Hence, a series of initiatives were started to improve clinical trial designs. The focus is indeed to move from large trials to smaller studies that enroll similar patients so that the treatment effect can be adequately measured [26]. Biomarkers play a central role in this context (Figure 2), being useful to enrich clinical trial CKD populations through at least three important ways called biomarker-based approaches: (1) by identifying patients at increased risk for developing an event (risk-based enrichment); (2) by selecting a population based on the response to a drug of interest (predictive response enrichment or adaptive enrichment); (3) by detecting subgroup of similar patients within a master trial protocol [136].

Risk-based enrichment was used in the proteomic prediction and renin angiotensin aldosterone system inhibition prevention of early diabetic nephropathy in type 2 diabetic patients with normoalbuminuria (PRIORITY) study. The PRIORITY study enrolled patients with diabetes mellitus and normal albuminuria at increased risk for developing albuminuria [137]. High or low risk was established based on urine CKD273 levels, and only high-risk patients were then randomized to receive spironolactone or placebo. Although the trials did not show a significant effect of spironolactone on preventing the development of albuminuria, high-risk patients identified with CKD273 were at increased risk of CKD progression vs. low risk patients (*p* < 0.001). PRIORITY was an innovative design, since it anticipated the treatment of albuminuria in patients who were only likely to develop albuminuria, but not yet with albuminuria. The adaptive enrichment design consists of exposing all patients to a short-term period (usually called run-in) of treatment with the drug of interest before randomization. In this case, biomarkers could inform on the response/non-response to treatment. Such a design was adopted in previous trials like the Study of Heart and Renal Protection (SHARP) study and more recently in the study of diabetic nephropathy with the endothelin receptor antagonist atrasentan (SONAR) trial [94,138]. Patients enrolled in SONAR underwent a six-month treatment period with atrasentan, and only patients who manifested a 30% reduction in albuminuria levels (measured as albumin-to-creatinine ratio) were then randomized. Hence, albuminuria worked as a biomarker for the prediction of response to treatment. Moreover, the SONAR trial included the assessment of “secondary” risk markers, such as the B-type natriuretic peptide levels. Patients who showed a significant increase in this marker during run-in were excluded from the study. This strategy allows assessing the individual response to treatment, including the effect of a drug on primary and secondary markers and, thus, to capture in a reliable manner the effect of treatment after randomization. An extension of the adaptive enrichment trial is given by the master trial protocol [139]. Master protocols can be planned to test the efficacy of multiple interventions, each targeting a subgroup of patients defined by a biomarker. Master protocols encompass umbrella, basket, and platform trials. Platform trials aroused the interest of the nephrology community [136]. The platform is an experimental cohort of patients followed periodically to assess laboratory and clinical measurements. Within the platform, multiple treatments can be started or withdrawn and, if a defined treatment shows benefits in a defined subgroup of the platform, it can be introduced in clinical practice [140]. This approach allows the acceleration of the experimental phase of drug development, to improve the application of biomarkers and to save time and financial sources. Future perspectives around the implementation of available biomarkers are depicted in Table 3. 

## 7. Conclusions

CKD is a growing public health problem with high morbidity and mortality. The current classification considers the eGFR and albuminuria levels to classify patients in prognostic categories. Novel biomarkers were also developed to improve risk stratification and clinical decision-making, as well as guide CKD patient enrichment in clinical trials. Despite the great efforts made, only a few biomarkers found a large clinical application to date. More emphasis should be placed on the development process of biomarkers, which needs to be methodologically rigorous, well validated, and correctly diffused. A “real-world” assessment of biomarkers that can be performed by analyzing large databases with long-term follow-up may substantially contribute to understanding whether a definite biomarker can find clinical application. The implementation of biomarkers in CKD is highly expected in the future, since they provide information on the mechanisms of kidney disease, improve clinical practice, and, in most cases, are able to forecast both CV and renal endpoints, which represent the most frequent events in CKD patients.

## Figures and Tables

**Figure 1 ijms-21-05846-f001:**
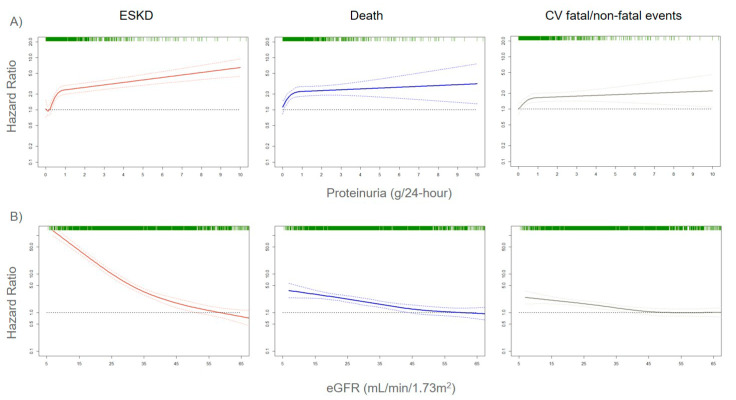
Adjusted risks for end-stage kidney disease (ESKD), death, and cardiovascular (CV) fatal and non-fatal events, by 24-h proteinuria (panel **A**) or estimated glomerular filtration rate (eGFR) (panel **B**) levels. Solid lines represent hazard ratios, whereas dashed lines the 95% confidence intervals. Hazard ratios were modeled by means of restricted cubic spline (RCS) due to the non-linear association with the endpoints. Knots are located at the zeroth, 25th, 50th, and 75th percentiles for proteinuria and 15, 30, 45, 60 mL/min/1.73 m^2^ for eGFR. Risks are adjusted for the four-variable Tangri equation [26]: age, gender, eGFR, and proteinuria. Rug plots on the *x*-axis at the top (colored green) represent the distribution of observations. Data source: pooled analysis of six cohorts of CKD patients referred to Italian nephrology clinics [27].

**Figure 2 ijms-21-05846-f002:**
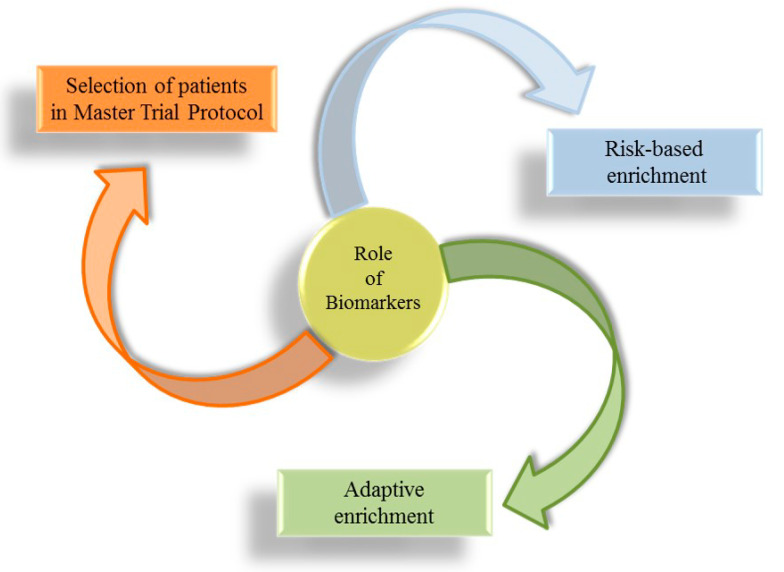
Biomarker-based approaches for patient selection in clinical trials.

**Table 1 ijms-21-05846-t001:** Summary of the principal prognostic and predictive biomarkers in chronic kidney disease patients.

Biomarkers	Characteristics	Prognostic/Predictive values
Cystatin C	Low-molecular-weight protein, produced by all types of nucleated cells, which acts as a cysteine protease inhibitor. It is freely filtered by the renal glomeruli, then 99% reabsorbed and metabolized in the renal proximal tubule; it is not secreted.	Cystatin C improves the estimation of eGFR and risk prediction of CV and renal events. It also allows a more precise stratification of patients according to their CV and renal risk [64].
β2-microglobulin	Protein present on the surface of immune cells, as a constant subunit of class I histocompatibility antigens. It is also found in blood and other biological fluids, as an expression of cell turnover. It is filtered by the renal glomerulus and reabsorbed at the tubular level.	It improves the prediction of ESKD, all-cause mortality, and new onset of CV disease [65].
hs-cTnT	Cardiac troponins are enzymes present in both skeletal and cardiac muscles. They regulate muscle contraction by controlling the calcium-mediated interaction of actin and myosin.	It improves the risk prediction of CV events, particularly heart failure regardless of the level of kidney function [55,57,114,115], as well as the risk prediction of microvascular events (nephropathy or retinopathy) in diabetic patients and the risk prediction of CKD progression.
NT-proBNP	Amino terminal fragment of the natriuretic type B peptide, normally produced in the heart and released in the case of cardiac stresses consequent to water overload conditions.	It improves the risk prediction of CV events, particularly heart failure regardless of the level of kidney function [55,57,114,115], as well as microvascular events (nephropathy or retinopathy) in diabetic patients and CKD progression. In the SONAR trial, it was used as a predictive biomarker in order to exclude patients with sodium retention after treatment with atrasentan [94]
sST2	A soluble form of the ST2 protein. It is a member of the interleukin 1 receptor family. In the case of myocardial stress, there is an upregulation of the ST2 gene and an increase in sST2 levels; by interacting with IL-33 (ligand for ST2), it counteracts the cardioprotective effect deriving from the ST2-IL 33 bond.	It showed an incremental prediction ability (over NT-proBNP) of death and hospitalizations due to HF in CKD patients [62,116]. It does not predict the risk of CKD progression.
GDF-15	Member of TGF-β cytokine family that is released in response to cellular stress. It appears to have a role in regulating inflammatory processes, apoptosis, cell repair, and cell growth.	It improves the risk prediction of both CV and microvascular events [62,116].
FGF-23	A protein belonging to the family of fibroblast growth factors, involved in the metabolism of phosphates. It is secreted in response to increased serum phosphate or calciprotein particles (colloidal nanoparticles of calcium phosphate dispersed in the blood), from osteocytes/osteoblasts. It acts on the kidneys by reducing the expression of a sodium phosphate transporter located in the renal proximal tubule, thus increasing the urinary excretion of phosphates.	It was significantly associated with mortality, atherosclerotic events, HF, and ESKD in CKD patients [52,53].
MMPs	Calcium-dependent endopeptidases that contain zinc and that are involved in the various processes of tissue development and cellular homeostasis.	Serum MMP-2, -8, and -9 and TIMP-1 are associated with atherogenesis, the severity of kidney damage, and the onset of left-ventricular hypertrophy and peripheral vascular disease [45,46,47,48,49,50,51]. MMPs levels are modified by selective and nonselective drugs. Changes in MMP levels are associated with a reduction of CV risk [102,103,104].
Urinary markers	Urinary markers of tubule damage (IL-18, KIM-1, NGAL), repair (YKL-40), and inflammation (MCP-1)	Increased urinary concentrations of these biomarkers predict a linear decline in eGFR over time [65,66,67].
eGFR_crea_	eGFR_crea_ is an estimation of the kidney function level, based on serum creatinine levels, age, gender, and race.	A reduction of eGFR is a potent predictor of CV and renal endpoints [21,22,23]. A treatment-induced reduction of eGFR (30% and 40% reduction) is considered to be a surrogate endpoint of ESKD [73,74,75].
Proteinuria	Presence of an abnormal quantity of proteins in urine. It is considered the principal marker of kidney damage.	The increase in proteinuria levels is strongly associated with the onset of fatal and non-fatal CV events [23,24,25]. In clinical trials, patients who developed a significant reduction in proteinuria levels, during the first months after treatment, were protected against CV and renal events over time [89,90,91,92,93,94,95,96,97].
F-Uprot	Proteinuria/eGFR × 100. It combines the prognostic/predictive power of two biomarkers (proteinuria and eGFR).	It improves risk stratification for ESKD outcomes at all the stages of CKD [27].
MPO	It is an enzyme belonging to the class of oxide reductase, with bactericidal and pro-inflammatory action.	It is a prognostic marker of cardiovascular risk, and it is associated with the risk of renal outcome (RRT, 50% eGFR decline, eGFR ≤ 15 mL/min/1.73 m^2^) [42,43,44].
RRI	Renal resistive index (RRI) is a ultrasonographic index of intrarenal arteries, defined as (peak systolic velocity − end diastolic velocity )/peak systolic velocity.	Raised RI levels above have been shown to reflect renal and systemic vascular impairment and predict CV events in hypertensive and CKD patients [105,106,107]. Medications as RAAS inhibitors and SGLT2-i reduce RRI levels over time and improve vascular damage [108,109].
ACE ID/DD	Insertion (I)/deletion (D) polymorphism of the angiotensin-converting enzyme (ACE) gene influences the circulating and renal activity of RAAS.	The D allele patients showed a poor CV prognosis in the RENAAL trial [110,111]. Patients with the DD genotype, despite being at a high risk of CV events, showed the better response to losartan in the RENAAL study [110,111].
Classifiers	A classifier is a combination of the informative markers able to classify patients according to their risk of developing an outcome or likelihood of response to a treatment.	13 metabolites predicted CKD progression in the CRIC cohort [76]. A panel of 21 metabolites was shown to predict the proteinuric response to ARBs [112].
CKD273	It is the combination of 273 urinary peptides identified as early indicators of molecular changes that predict the development or progression of CKD. The main components of CKD273 are collagen fragments and protein fragments, including proteins involved in inflammation.	It predicts the risk of development or progression of CKD, allowing the implementation of preventive attitudes. It is used in clinical trials to predict the development of CKD in response to a therapeutic approach [72,73,74,75].

eGFR, estimated Glomerular Filtration Rate; CV, Cardiovascular; ESKD, End-Stage-Kidney-Disease; hs-cTnT, high-sensitivity cardiac troponin; CKD, Chronic Kidney Disease; NT-proBNP, N-terminal pro-B-type natriuretic peptide; SONAR, study of diabetic nephropathy with the endothelin receptor antagonist atrasentan; sST2, soluble form of ST2; IL, interleukin; HF, Heart Failure; GDF-15, growth differentiation factor-15; TGF-β, transforming growth factor β; FGF-23, Fibroblast Growth Factor 23; MMP, Matrix metalloproteinases; TIMP, tissue inhibitor of metalloproteinases; KIM-1, Kidney Injury Molecule-1; NGAL, neutrophil gelatinase-associated lipocalin; YKL-40, repair human cartilage glycoprotein-40; MCP-1, monocyte chemoattractant protein-1; MPO, Myeloperoxidase; RRT, Renal Replacement Therapies; RAAS, Renin–Angiotensin–Aldosterone System; SGLT2-i, sodium–glucose cotransporter 2 inhibitors; RENAAL, Reduction of End Points in Non-Insulin-Dependent Diabetes with the Angiotensin II Antagonist Losartan; CRIC, Chronic Renal Insufficiency Cohort; ARBs angiotensin-receptor blockers.

**Table 2 ijms-21-05846-t002:** Principal tools used to assess analytic validation.

Features	Definition	Statistical Metric
Precision	Intra-assay agreement of a set of results among themselves. It could be expressed by coefficient of variation (CV).	CV_(%)_ = (Standard deviation_samples_/Mean_samples_) × 100
Reproducibility	Concordance between various measurements carried out in different laboratories and experimental conditions on the same sample.	
Accuracy	Closeness of the agreement between result of a single measurement and true value obtained using a reference standard method.	
Trueness	Concordance between a series of assays and the real value of analyte concentration.	
Bias	Systematic difference of the series of measurements with true value.	Bias_(%)_ = (Mean_sample_ − True value) × 100
Limit of blank	Highest apparent analyte concentration founded by testing specimens without analyte.	LoB = Mean_blank_ + 1.645(SD_blank_)
Limit of detection	Average of lowest concentration of analyte which can be distinguished from a blank sample.	LoD = LoB + 1645(SD_samples_)
Limit of quantification	Smallest concentration of analyte with an acceptable accuracy and precision.	
Linearity	Proportionality between a set of measured values and true concentration of analyte.	
Analytic specificity	Ability to measure only and exclusively the analyte of interest.	
Analytic sensibility	Ability to measure lowest concentration of analyte.	

**Table 3 ijms-21-05846-t003:** Validation score and future perspectives in the development of biomarkers in CKD patients.

Biomarkers	Validation Criteria	Future Perspectives
Proteinuria	Analytic validation: +/−	Further studies are needed to establish (1) how this marker should be used for monitoring disease progression considering, among all factors, its variability, and (2) what are the true cut-offs for response to treatments. Inclusion of proteinuria in risk prediction models that include the presence of renal diagnoses is also needed.
	Clinical proof of concept: +	
	Clinical prospective validation: +	
	Incremental value of the biomarker: +	
	Introduction in clinical trials: +	
eGFR_crea_	Analytic validation: +	eGFR_crea_ is an important marker used to stratify risk in CKD patients. Further studies could refine the assessment of eGFR_crea_ as a biomarker of response to nephroprotective treatments in clinical trials. Inclusion of eGFR_crea_ in risk prediction models that include the presence of renal diagnoses is also needed.
	Clinical proof of concept: +	
	Clinical prospective validation: +	
	Incremental value of the biomarker: +	
	Introduction in clinical trials: +/−	
Markers of oxidative stress, tissue remodeling, and metabolism	Analytic validation: +	The prognostic role of these markers should be evaluated in larger cohort studies. Individual prognostic measures should be provided. Although pilot experimental trials showed promising results, stronger evidence in CKD patients around the changes in these markers after treatment initiation is needed.
	Clinical proof of concept: +	
	Clinical prospective validation: +/−	
	Incremental value of the biomarker: −	
	Introduction in clinical trials: +/−	
Cardiac markers	Analytic validation: +/−	Although cardiac markers levels are associated with the severity of CKD, their assessment is confounded by the coexistence of CV disease, as well as by the eGFR levels. Further studies are needed to establish the true role of these markers in CKD patients.
	Clinical proof of concept: +	
	Clinical prospective validation: +/−	
	Incremental value of the biomarker: +/−	
	Introduction in clinical trials: −	
Filtration and urinary markers	Analytic validation: +/−	The prognostic role of these markers should be evaluated in larger studies. Individual risk prediction models that include these parameters and intervention studies assessing their changes over time should be implemented.
	Clinical proof of concept: +	
	Clinical prospective validation: +	
	Incremental value of the biomarker: −	
	Introduction in clinical trials: −	
Ultrasound markers	Analytic validation: +	RRI was found to be associated with CV and renal events in CKD patients, being a promising marker. However, larger clinical trials evaluating the association between changes (treatment-induced) in RRI and clinical outcomes should be performed in the future.
	Clinical proof of concept: +	
	Clinical prospective validation: +	
	Incremental value of the biomarker: +/−	
	Introduction in clinical trials: +/−	
Proteomics, metabolomics, and genomics	Analytic validation: +	Omics approaches show useful prognostic and predictive information in addition to traditional risk factors. Improving the inclusion of these markers in clinical trials may inform on their clinical applicability.
	Clinical proof of concept: +	
	Clinical prospective validation: +	
	Incremental value of the biomarker: +	
	Introduction in clinical trials: +/−	

+, fully present; +/−, partially present; −, absent. CKD, chronic kidney disease; eGFR, estimated glomerular filtration rate; RRI, renal resistive index; CV, cardiovascular.

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
