# Peer review of "Contribution of Predictive and Prognostic Biomarkers to Clinical Research on Chronic Kidney Disease"

_ijms, 2020, doi:10.3390/ijms21165846_

Round 1

Reviewer 1 Report

This is an extensive review on a great variety of biomarkers in CKD. I have added several comments directly in the manuscript attached.

Generally, the role of several suggested biomarkers remains unclear at this time. Also, some markers may be biased e.g. cardiorenal parameters or influence of (e.g. intermittent) hypertension on the quantity of proteinuria.

Extrapolations from some papers seem problematic in their general significance, e.g. in the SPRINT Trial CV deaths are 37 vs 155 from any cause. Yet it was not clarified, where to find SCD (which would be the majority of CKD deaths and ought to be a CV death as well; so these data are rather difficult to interpret, which makes evaluation of biomarkers even more difficult). Similarly, application of NT-pro-BNP or TropT may have two competing causes, or a combination of both. So the finding of an association in a collective does not essentially help in the risk estimation of the individual (which clinicians need to do and biomarkers promise to perform).

For clinicians this is a tough read, even more, since many markers remain unestablished. So, perhaps it would be possible to categorise the markers in the fashion described in the manuscript. Up to now it seems rather difficult to select among the new biomarkers for clinical studies, since several have not been characterised well enough for solid interpretation from my point of view.

Furthermore it might be beneficial to try some way of further stratification of biomarker portrayal - the single flow of text is hard to read. 

Author Response

This is an extensive review on a great variety of biomarkers in CKD. I have added several comments directly in the manuscript attached.

R. We thank the Reviewer for having carefully revised our manuscript.

Following his comments, we made several changes throughout the manuscript that, when considered as a whole, have helped us to improve significantly our paper in terms of readability and completeness of its contents. We also had the opportunity to clarify the points that were less clear or incomplete in the previous version of the text.

We have answered to the comments that the Reviewer added directly to the manuscript (please see below ‘Replies to other Reviewer 1 comments’) and to those added as ‘comments to the Authors’. Furthermore, we have amended the multiple ‘spelling’ errors that were likely due to an inconsistency between the Word format files  “.doc” and “.docx”.

Generally, the role of several suggested biomarkers remains unclear at this time. Also, some markers may be biased e.g. cardiorenal parameters or influence of (e.g. intermittent) hypertension on the quantity of proteinuria.

R. Thank you for raising this point . We have now extensively revised the central part of the manuscript focused on the role of each biomarker. With respect to albuminuria, we consider it important to discuss the strong association between the blood pressure (as well as blood pressure variability) and albuminuria levels. Hence, we have expanded this point in the manuscript (please see Section “Prognostic biomarkers in CKD- Kidney biomarkers” page 5). Similarly, we have revised the concepts regarding the prognostic role of cardiac markers by examining with more attention the suggested Literature on the topic (please see Section “Prognostic biomarkers in CKD-Cardiac biomarkers” pages 6-7). Such implementations have helped us to present a less static (and hopefully more critical) discussion on the biomarkers, which is also more consistent with the introduction and conclusion of our article. We have also highlighted in this revised version that albuminuria and eGFR have been included in risk prediction models with the aim to assess an individual risk prediction (PMID:PMC4594193, DOI: 10.2215/CJN.01290216, DOI:10.1016/j.kint.2018.08.009, doi: 10.1001/jama.2011.451).

Extrapolations from some papers seem problematic in their general significance, e.g. in the SPRINT Trial CV deaths are 37 vs 155 from any cause. Yet it was not clarified, where to find SCD (which would be the majority of CKD deaths and ought to be a CV death as well; so these data are rather difficult to interpret, which makes evaluation of biomarkers even more difficult). Similarly, application of NT-pro-BNP or TropT may have two competing causes, or a combination of both. So the finding of an association in a collective does not essentially help in the risk estimation of the individual (which clinicians need to do and biomarkers promise to perform).

R. We thank the Reviewer for his comment. We have now revised the discussion about cardiac biomarkers in CKD patients (please see section “Prognostic biomarkers in CKD-Cardiac biomarkers” pages 6-7). In particular, we have mentioned the fact that most cardiac biomarkers are unable to discern from CV and renal dysfunction. With respect of the SPRINT trial, we have mentioned the renal endpoint of the trial (composite outcome of 50% eGFR decline or ESKD requiring dialysis or kidney transplantation and eGFR decline), but not the CV endpoint. As regards Sudden Cardiac Death, we have now clarified what were the CV endpoints measured in large cited cohort studies (CV mortality, Coronary Heart Disease, Stroke, Heart Failure). Please see manuscript page 3. At the same time, we have now expanded the discussion by including Sudden Cardiac Death (SCD) among CV endpoints. We have mentioned the important role of eGFR reduction (both estimated through serum creatinine or cystatin C) in SCD risk prediction. Please, see manuscript Section “Prognostic biomarkers in CKD-kidney biomarkers” pages 4, 5 and “Filtration and urinary biomarkers” page 7.

For clinicians this is a tough read, even more, since many markers remain unestablished. So, perhaps it would be possible to categorise the markers in the fashion described in the manuscript. Up to now it seems rather difficult to select among the new biomarkers for clinical studies, since several have not been characterised well enough for solid interpretation from my point of view.Furthermore it might be beneficial to try some way of further stratification of biomarker portrayal - the single flow of text is hard to read.

R. We agree. In fact, the classification of biomarkers into prognostic and predictive could be considered a methodological classification. Biomarkers are also commonly classified according to the site of origin/mechanism of action (doi:10.1161/CIR.0000000000000664) as well as to the outcome investigated (i.e. markers of cardiovascular or renal risk). We agree with the reviewer that the classification into prognostic and predictive is hard to follow, since too much information is presented together. We have now tried to simplify the two paragraphs (prognostic and predictive biomarkers) by introducing a further - site of origin – type classification.

Replies to other Reviewer 1 comments

  • We have added “by” before “41.5%”. Please see Introduction, page 2.

Originally it was intended to use a "CGA" classification (cause, eGFR, albuminuria) to better adress the importance of underlying disease. Clearly, it is not possible to find a unifying risk estimation with various diseases of very diverging activity and aggression. There is much debate, whether in old age a diminution of eGFR without apparent disease might be a "disease" or result of old age (physiological ageing)

R. This is a good suggestion. Accordingly, we preferred now to mention the CGA classification directly in the article.We clarified that the idea of CGA classification is correct, but the risk estimates that also include renal diagnoses are still controversial mainly because of a different categorization of causes among cohort and prognostic studies (such as those documented in references #12 and #13). We have also mentioned the debate around the role of eGFR in the elderly and, since we considered them important additions to the manuscript, we mentioned some evidence derived by the pro-con “doi:10.1093/ndt/gft324 vs. doi:10.1093/ndt/gft306”. Please, see manuscript section “Introduction” at page 2.

Albuminuria may further be influenced by level of blood pressure at the random time of measurement, which may bias risk estimation. There is a general problem of trying to perform risk predicitions from a one-point-measurement (which in some cases, biomarkers are suggested to do). Perhaps authors should address this aspect.

R.We fully agree with this comment. Blood pressure is another important variable that influences albuminuria levels, particularly in patients with lower eGFR. We have discussed these concepts in the Section “Prognostic biomarkers in CKD” (please see manuscript page 5). The limitation of a single measurement is also important (and it is true for albuminuria as well as other biomarkers). In fact, the International Society of Nephrology (ISN) noted that a consensus on how often albuminuria should be measured over time to correctly assess CV/renal risk prediction and to monitor the course of Chronic Kidney Disease, is still lacking. We also mentioned this unmet need in the article.

There is a further problem in course of this manuscript: in some places, creatinine or eGFr are addressed as biomarkers - yet at the same time they serve as scale of measurement to determine the quality of additional biomarkers - this is a discrepancy in itself.

R. We have clarified this point in the section “Prognostic biomarkers in CKD” (page 5). Indeed, proteinuria and eGFR are strongly and independently (from each other as well as from other comorbidities) associated with a worse prognosis in CKD patients. However, taken together these kidney measures did not provide a full and complete risk prediction. Hence, despite the fact that their assessment is essential to diagnose and monitor CKD, novel biomarkers should demonstrate to provide useful prognostic information not only beyond, but more appropriately‘beside and beyond’ proteinuria and eGFR. We have refined this part in the manuscript.

Srisravasdi Am J Clin Path 2010; co-influence of heart and kidney on biomarkers. There is a bias and a problem: in many cases, patients suffer from damages or diseases to both organs, heart and kidney to various extent. Biomarkers cannot discern these dual influences. Furthermore, clearance of these markers depends on eGFR. Thus, introduction of theses biomarkers to determine prognosis (which should address one problem, not two) seems rather problematic.

R. We have toned-down the role of CV biomarkers by reporting that the best application of the cardiac biomarkers is currently focused on patients with cardiorenal syndromes (doi:10.1161/CIR.0000000000000664), a great example of the strict linkage between the heart and the kidney. Moreover, we have also revised the prognostic studies (both on CV and renal risk), which presented controversial results on the association between cardiac markers and outcomes, in CKD patients. Please, see Section “Prognostic biomarkers in CKD-cardiac biomarkers” pages 6-7.

As mentionned previously, it is to some extent problematic to vary scales, when assessing other parameters.Risk estimation without consideration of underlying diseases creates a problem of its own, at least from a clinicians point of view. These estimations do not help clinicians or patients, at least from my point of view, since individual risk estimation is impossible.

R. We agree with this point. In fact, even if risk prediction models have been already published, in most cases they do not encompass the risk associated with underlying renal causes in addition to traditional and non-traditional risk factors. We have mentioned this problem in the section “Prognostic biomarkers in CKD” at page 4.

  • We have specified that BRCA1/2 are ‘causative’ mutations, making it possible for their role as predictive biomarkers in Oncology. Section “Predictive biomarkers in CKD”Page 9.
  • We have replaced “These” with “This” at page 9, as suggested by the Reviewer.
  • Following the reviewer’s suggestion (and since we consider this appropriate), we have integrated the sentence “This means that if a treatment is started on the basis of a blood/urine biomarker level, the individual prognosis may remain unchanged or even worsen, due to the presence of other active mechanisms of damage..” by adding “as well as - most important - in different disease entities, that cause the chronic decline of renal function through diverse pathophysiological pathways”. Please see page 9.
  • We have added “been” in the sentence “RRI has also been found to predict..” on page 10.
  • With respect to the interaction between Losartan effect and ACE/DD-/ACE-ID polymorphisms we have clarified that the best losartan effect have been found in patients who needed substantial nephoprotection, namely D allele carriers. Please see manuscript section “Predictive biomarkers in CKD”page 10.
  • Typos and mistakes in tables 1 and 2 have been amended.

As far as I could comprehend, most of the new biomarkers have up to date failed to contribute to this task. perhaps it would be helpful to categorise the view of the authors, which of these markers might be promising in the future, possibly using a table, which uses the classification criteria mentioned on page 10?

R. Yes indeed. As mentioned by the recent report of International Society of Nephrology (ISN), a great part of biomarkers did not reach the phase of clinical application, and in some cases also that of a rigorous validation (DOI: 10.3390/medicina55060268). For example, many inflammatory/tissue remodeling markers, albeit providing interesting prognostic and predictive information, have not been validated yet in large cohorts of patients with individual measures (i.e. discrimination, calibration, reclassification, validation). Conversely, cardiac markers have shown only a little improvement in clinical utility when these measures have been tested. Based on these points, the ISN prompted the creation of large databases (multicenter and multinational) from which it will be possible to determine the “real world” of each biomarker, that is, whether a biomarker is useful and should be integrated in clinical practice. We have now added a short sentence in the Conclusions (page 20) and a Table 3 which depicts a score (based on the validation parameters presented at page 10) of the development of available biomarkers and the future perspectives.

Reviewer 2 Report

Provenazo et al. have written an exquisite review on the contribution of predictive and prognostic biomarkers to clinical research into CKD. The authors are clear to define terms and later detail specific examples of both prognostic and predictive biomarkers using data from relevant studies. The review well organised, interesting and of high quality. The subject area is of huge clinical and research value and I will be recommending this article for publication.

Just a few minor comments:

  • Issues with spacing in document – many words are concatenated together - perhaps an issue with pdf format?
  • Couple of typos to fix
  • On page 7, the authors briefly discussed the polymorphism (ACE/ID) and how it was able to predict the response to losartan. Can I suggest the authors also consider polymorphisms that can be used as prognostic biomarkers or at least consider these? Plenty of literature emerging that shows how genetic predisposition in known renal genes influences prognosis of CKD. Could also add in a sentence about projects such as UK Biobank.

Author Response

Provenazo et al. have written an exquisite review on the contribution of predictive and prognostic biomarkers to clinical research into CKD. The authors are clear to define terms and later detail specific examples of both prognostic and predictive biomarkers using data from relevant studies. The review well organised, interesting and of high quality. The subject area is of huge clinical and research value and I will be recommending this article for publication.

R. We would like to thank the Reviewer for the positive and encouraging comment. Indeed, this is an important topic and the purpose of this article is to provide helpful elements on how to improve the great work of the past on biomarkers development and validation in the CKD population.

Just a few minor comments:

Issues with spacing in document – many words are concatenated together - perhaps an issue with pdf format?

R. Yes indeed. We have double checked this format issue and we think it is likely due to an inconsistency between the word format “.doc” and “.docx”. However, we have now amended all the concatenated words before resubmitting the manuscript.

Couple of typos to fix

R: We have checked typos throughout the manuscript and fixed them.

On page 7, the authors briefly discussed the polymorphism (ACE/ID) and how it was able to predict the response to losartan. Can I suggest the authors also consider polymorphisms that can be used as prognostic biomarkers or at least consider these? Plenty of literature emerging that shows how genetic predisposition in known renal genes influences prognosis of CKD.

R. This is a very important comment. We have implemented the prognostic section by describing the prognostic role of polymorphisms and genetics in CKD. In particular, we have first mentioned the overall prevalence of genetic causes of CKD and then described the prognostic role of the major single-nucleotide polymorphisms (SNPs) in CKD patients including the genes UMOD and APOL1 encoding respectively for uromodulin and apolipoprotein L1, as well as the SNPs: PRKAG2, LASS2, DAB2, DACH1 and STC1. We have also mentioned the genetic risk score built using data of the PREVEND study, which showed how the inclusion of combined SNPs information may improve risk prediction (and thus prognostic estimates) in CKD patients. All these evidences encourage the use of genetic measurements in clinical practice early in the future. Please, see manuscript section “Prognostic biomarkers in CKD” page 8.

Could also add in a sentence about projects such as UK Biobank.

R: We have included (page 8) a statement about the great work developed on the UK biobank. In particular, we mentioned the UK biobank design (doi: 10.1371/journal.pmed.1001779), the association between genetically predicted testosterone (doi:10.1186/s12916-020-01594-x) and insulin resistance (doi:10.1007/s00125-020-05163-y) with CKD and kidney function and the important finding, as albuminuria is a widely used biomarker, that genetic pathways of urine albumin-to-creatinine ratio are multiple and are able to predict cardiovascular events (doi:10.1093/hmg/ddz243).

Round 2

Reviewer 1 Report

The revisions have greatly improved the manuscript